# The Mechanism of Honey in Reversing Metabolic Syndrome

**DOI:** 10.3390/molecules26040808

**Published:** 2021-02-04

**Authors:** Khairun-Nisa Hashim, Kok-Yong Chin, Fairus Ahmad

**Affiliations:** 1Department of Anatomy, Faculty of Medicine, Universiti Kebangsaan Malaysia, Jalan Yaacob Latif, Bandar Tun Razak, Kuala Lumpur 56000, Malaysia; khairunnisabthashim@gmail.com; 2Department of Pharmacology, Faculty of Medicine, Universiti Kebangsaan Malaysia, Jalan Yaacob Latif, Bandar Tun Razak, Kuala Lumpur 56000, Malaysia; chinkokyong@ppukm.ukm.edu.my

**Keywords:** metabolic syndrome, honey, obesity, hyperglycaemia, hyperlipidaemia, obesity, hypertension

## Abstract

Metabolic syndrome is a constellation of five risk factors comprising central obesity, hyperglycaemia, dyslipidaemia, and hypertension, which predispose a person to cardiometabolic diseases. Many studies reported the beneficial effects of honey in reversing metabolic syndrome through its antiobesity, hypoglycaemic, hypolipidaemic, and hypotensive actions. This review aims to provide an overview of the mechanism of honey in reversing metabolic syndrome. The therapeutic effects of honey largely depend on the antioxidant and anti-inflammatory properties of its polyphenol and flavonoid contents. Polyphenols, such as caffeic acid, p-coumaric acid, and gallic acid, are some of the phenolic acids known to have antiobesity and antihyperlipidaemic properties. They could inhibit the gene expression of sterol regulatory element-binding transcription factor 1 and its target lipogenic enzyme, fatty acid synthase (FAS). Meanwhile, caffeic acid and quercetin in honey are also known to reduce body weight and fat mass. In addition, fructooligosaccharides in honey are also known to alter lipid metabolism by reducing FAS activity. The fructose and phenolic acids might contribute to the hypoglycaemic properties of honey through the phosphatidylinositol 3-kinase/protein kinase B insulin signalling pathway. Honey can increase the expression of Akt and decrease the expression of nuclear factor-kappa B. Quercetin, a component of honey, can improve vasodilation by enhancing nitric oxide production via endothelial nitric oxide synthase and stimulate calcium-activated potassium channels. In conclusion, honey can be used as a functional food or adjuvant therapy to prevent and manage metabolic syndrome.

## 1. Introduction

Metabolic syndrome is a constellation of five risk factors that predispose an individual to type 2 diabetes (T2DM) and cardiovascular disease (CVD) [1]. Previously known as “Syndrome X” by Reaven, metabolic syndrome was described as the tendency of glucose intolerance, hyperinsulinemia, hypertension, and dyslipidaemia occurring simultaneously rather than by chance alone [2]. The idea of introducing a diagnostic criterion for metabolic syndrome was first attempted by the World Health Organization (WHO) in 1999, which includes insulin resistance and glucose intolerance in addition to other components, such as elevated blood pressure, dyslipidaemia, obesity (which is determined by waist/hip ratio, or body mass index (BMI)), and microalbuminuria, as a diagnosis of metabolic syndrome. Although the definitions by Reaven and the WHO are more hyperglycaemia centric, other organizations, such as the International Diabetes Federation (IDF), American Heart Association (AHA), and the National Heart Lung and Blood Institute (NHLBI), have also put forth their classification and diagnostic criteria, which has a more neutral approach. The most recent classification for the diagnosis of metabolic syndrome, known as the Joint Interim Statement (JIS), was proposed to unify the previous criteria [1,3]. The JIS criteria for the clinical diagnosis of metabolic syndrome include:Elevated fasting blood glucose (FBG) ≥ 7.0 mmol/L (or drug treatment for elevated glucose).Elevated blood pressure (systolic blood pressure (SBP) ≥ 130 mmHg and/or diastolic blood pressure (DBP) ≥ 85 mmHg) (or drug treatment for hypertension).Elevated serum triglyceride (TG) ≥ 1.7 mmol/L (or drug treatment for elevated TG).Reduced high-density lipoprotein (HDL-C) (1.0 mmol/L in male and 1.3 mmol/L in female) (or drug treatment for reduced HDL-C).Elevated waist circumference (WC) (according to population and country-specific definitions).

According to the IDF consensus on the worldwide definition of metabolic syndrome, the worldwide prevalence of metabolic syndrome in the adult population is estimated to be 20–25% [4]. It is estimated that around 12–37% of the Asian population and 12–26% of the European population suffer from metabolic syndrome [5]. In other words, about 25% of the world population is affected by this condition [6]. A systematic review of the prevalence of metabolic syndrome in the Asia Pacific demonstrated that close to 20% of the adult populations were affected by metabolic syndrome [5]. Malaysia has the highest metabolic syndrome prevalence at 34.3% compared to other South East-Asian countries, such as Indonesia (28.9%), the Philippines (11.9%), and Singapore (20.2%) [7].

Lifestyle changes continue to be the primary approach in treating metabolic syndrome, while drug therapy focuses on treating each metabolic syndrome component [8]. Often, patients present multiple conditions and require multiple drugs, resulting in polypharmacy [3]. Due to the increasing metabolic syndrome prevalence, polypharmacy increases the financial burdens to the national healthcare system and the patients themselves [9]. In addition, non-compliance, side effects of medications, drug-to-drug interactions, and numerous visits to physicians can escalate national and personal healthcare burdens. Therefore, prevention is far more economical in managing the epidemic of metabolic syndrome [3].

In recent years, various studies have demonstrated the use of natural ingredients, such as cinnamon, ginger, and other plant-derived therapeutics, as a dietary intervention in preventing or treating metabolic syndrome [10,11,12]. Some of the apitherapy agents intensively investigated include honey, propolis, pollen, bee venom, and royal jelly [13]. Honey is speculated to be a potent metabolic syndrome preventive agent due to its antioxidant, anti-inflammatory [14], hepatoprotective [15], antihypertensive [16], and antiobesity properties [17].

Honey is a natural by-product from the flower nectar and aerodigestive tract of honey bees, which contains various complex biochemical components. Fructose (36%) and glucose (31%) are the main carbohydrate constituents of honey. Other constituents found within honey include mineral, proteins, vitamin, organic acids, flavonoids, phenolic acid, and enzymes [18]. The antioxidant property of honey is strongly correlated with its phenolic content and colour intensity [19]. Due to its exceptional medicinal value, honey has been used widely in alternative therapy.

The purpose of the review is to provide an overview of the mechanism of honey in preventing or managing metabolic syndrome. A better understanding of the antimetabolic properties of honey will facilitate its incorporation in the management of metabolic syndrome.

## 2. Pathophysiology of Metabolic Syndrome

Metabolic syndrome can be contributed by various factors, such as diet, physical inactivity, and genetic predisposition [20]. In the following section, the pathogenesis of each component of metabolic syndrome is discussed.

### 2.1. Obesity

Obesity is one of the main components of metabolic syndrome. According to the JIS criteria, central adiposity is determined by waist circumference. The cut-off values for central obesity based on waist circumference differ by country. For example, they are 102 cm (40 inches) in Caucasian men and 88 cm (34.6 inches) for Caucasian women. Meanwhile, the values are 90 cm (35 inches) in Asian men and 80 cm (32 inches) in Asian women [1]. Abdominal obesity is principally caused by increased consumption of calorie-dense food (energy intake) as well as reduced physical activity (energy expenditure), leading to the formation of adipose tissue as a storage for the excessive energy [3].

Obesity is closely related to various chronic diseases, such as CVD, T2DM, non-alcoholic fatty liver disease (NAFLD), and cancer. At the same time, it is also associated with a state of low-grade inflammation. Inadequate blood supply to adipose tissue can lead to hypoxia, causing the release of free fatty acid (FFA), plasminogen activator inhibitor-1 (PAI-1), and cytokines, such as tumour necrosis factor-alpha (TNFα), interleukin-1-beta (IL-1β), and interleukin-6 (IL-6). The alteration in FFA release results in the accumulation of lipids and ectopic fat and impairs insulin sensitivity when exposed to other insulin-sensitive tissues, such as skeletal muscle. Furthermore, the cytokines further activate the inflammatory program, therefore aggravating inflammation and insulin resistance [21]. Meanwhile, PAI-1 is found abundantly in abdominally obese subjects and exerts its effect by inhibiting tissue plasminogen activator (tPA); therefore, it increases the risk of cardiovascular events [3]. Besides this, adipose tissue also secretes adipokines, such as leptin, adiponectin, and resistin [22]. Leptin overstimulation can lead to resistance, therefore thwarting satiety. Subsequently, it will affect glucose homeostasis, pancreatic ß cell functions, and other insulin-sensitive tissues [3,20,23]. Adiponectin acts as a protective protein hormone, which regulates lipid and glucose metabolism and increases insulin sensitivity by increasing glucose transport in the muscle and enhancing fatty acid oxidation [3]. Resistin shares a similar structure to adiponectin, but it promotes insulin resistance by hepatic gluconeogenesis. Its expression is three times higher in preadipocytes than mature adipocytes, suggesting its potential role in adipogenesis [22].

### 2.2. Hyperglycaemia

Insulin resistance is the impairment of glucose metabolism marked by an abnormal response to a glucose challenge. Previously, Reavan has referred to metabolic syndrome as “insulin resistance syndrome” because he believes that it is the unifying mechanism leading to the development of metabolic disorders [2,24]. Nutrient-induced toxicity due to overnutrition can lead to insulin resistance in tissues like skeletal muscle and heart tissues, which normally respond to insulin for glucose uptake. Insulin resistance is an adaptive mechanism of the tissues to avoid damage due to nutrient overload [25]. Adipokines plays an important role in fuelling insulin resistance. High FFA levels could cause a failure to limit glucose entry into cells, thus resulting in glucolipotoxicity [26]. Similarly, cytokines like TNFα or IL-1β promote insulin resistance by inhibiting insulin receptor substrate-1 (IRS-1), which plays a role in transmitting signals from insulin to intracellular pathways, such as the phosphoinositide 3-kinase/protein kinase B (PI3k/Akt) and extracellular signal-regulated kinase/mitogen-activated protein kinase (Erk/MAPK) pathway, leading to a reduction in glucose uptake. This event will consequently increase fasting glucose and reduce insulin-mediated glucose clearance. In this case, insulin cannot produce a normal insulin response in the target tissues, causing the β cells to produce more insulin, leading to hyperinsulinaemia. Over time, the lack of insulin production eventually leads to an inability to correct insulin resistance, which inadvertently gives rise to hyperglycaemia and T2DM [3]. Adiponectin is an anti-inflammatory adipokine, which enhances glucose transport in muscles and improves insulin sensitivity. The adiponectin level is conversely related to insulin resistance [27,28].

With metabolic syndrome, insulin resistance may contribute to the pathogenesis of diseases, such as NAFLD, polycystic ovary syndrome, T2DM, and atherosclerotic cardiovascular diseases [29,30,31,32].

### 2.3. Dyslipidaemia

Dyslipidaemia is characterized by reduced HDL-C, as well as increased very-low-density lipoprotein cholesterol (VLDL-C), TG, and low-density lipoprotein cholesterol (LDL-C) levels. This atherogenic lipid composition is the key component of the CVD risk factor, especially in T2DM individuals [33]. The JIS criteria consider reduced HDL-C and elevated TG for metabolic syndrome [1]. The changes in lipid metabolism and release are associated with FFA flux secondary to insulin resistance. The flux of FFA released from insulin-resistant tissues into the liver in a state of adequate glycogen will promote TG, apolipoprotein B (apo B) and VLDL-C production. The ability of insulin to impede FFA release leads to enhanced hepatic VLDL-C production, which may progress to fatty liver [34,35].

On the other hand, high VLDL-C and TG results in lower HDL-C and increased LDL-C levels because VLDL-transported TG is exchanged for HDL-transported cholesteryl ester (CE) via cholesteryl ester transport protein (CETP). Therefore, there is a high amount of atherogenic cholesterol-rich VLDL particles and TG-rich but cholesterol-depleted HDL particles in circulation. TG-rich HDL is hydrolyzed by the enzymes hepatic lipase or lipoprotein lipase, and HDL remnants are washed out from the systemic circulation, resulting in low HDL-C levels. Meanwhile, high-level VLDL-TG also enables CETP to promote the transfer of TG into LDL in exchange for LDL-transported CE. The TG-rich LDL undergoes hydrolysis by hepatic lipase, leaving a small dense LDL remnant [35]. The small dense LDL is a powerful risk factor for CVD, as it is considered highly atherogenic. It is easily oxidized and binds more readily to proteoglycan in the arterial wall [33].

Another important source of TG and VLDL production is via de novo fatty acid (FA) synthesis or lipogenesis [34]. Horton et al. showed that hepatic lipogenesis is regulated by the transcription factor sterol regulatory element-binding protein-1-c (SREBP-1c) [36]. The expression of SREBP is augmented by insulin in insulin target tissues, such as liver, skeletal muscle, and fat tissues [37,38]. SREBP binds to sterol regulatory element (SRE) found in multiple genes, subsequently activating cascades of enzymes involved in biosynthesis pathways, such as 3-hydroxy-3-methyl-glutaryl CoA (HMG-CoA) reductase and fatty acid synthase (FAS) [39]. FAS is an important subsequential component of lipid synthesis, which mainly functions to catalyze the synthesis of palmitate from acetyl-CoA and malonyl-CoA [40]. In rat studies, dietary carbohydrate (fructose) can increase the transcriptional rates of FAS by enhancing the stability of FAS mRNA in the liver [41,42].

Therefore, the proatherogenic lipid composition present in dyslipidaemia contributes to metabolic syndrome-induced cardiovascular disease risk.

### 2.4. Hypertension

Hypertension is an elevation in blood pressure. Approximately 80% of individuals with metabolic syndrome suffer from hypertension [43]. The AHA guideline defines hypertension as SBP > 130 mmHg and DBP > 80 mmHg [44]. Studies indicated that excess adiposity might play an important role in the pathogenesis of hypertension. As evidence, 65–75% of the risk factor for primary hypertension is contributed by obesity and excess weight gain. Visceral adiposity may result in a mild to moderate increase in several components of the renin-angiotensin-aldosterone system (RAAS), which includes angiotensin II (AT-II) and aldosterone [45]. Apart from this, obesity may also increase Ras-related C3 botulinum toxin substrate 1 (Rac 1), a guanosine triphosphate (GTP) binding protein that stimulates mineralocorticoid (MR) receptor, causing reabsorption of sodium, and subsequently increasing the intracellular volume, thus leading to an elevation in blood pressure [46,47].

Insulin resistance has also been linked to hypertension because insulin can cross the blood-brain barrier (BBB) to the central nervous system (CNS) and activate the systemic nervous system (SNS). Insulin also upregulates AT-II receptor and reduces nitric oxide (NO), which incites peripheral vascular resistance. Therefore, blockage of AT-II receptor, angiotensin-converting enzyme (ACE), or MR receptors may attenuate sodium retention, volume expansion, and elevation of blood pressure [43,44].

The chronic inflammation and oxidative stress that occur in metabolic syndrome could also contribute to endothelial dysfunction, which is the primary cause of many cardiovascular diseases, including hypertension. Diminished production/bioavailability of and response to NO, a proinflammatory state, and an imbalance between relaxing/contracting factors of the endothelial due to oxidative damage are postulated to be the major contributing factors of endothelial dysfunction. These changes will progress to alternation in vascular tone, phenotypic and microstructural changes of the vascular epithelial, which results in hypertension [48].

### 2.5. The Role of Oxidative Stress and Inflammation in Metabolic Syndrome

Metabolic syndrome is closely linked with oxidative stress, which occurs when reactive oxygen species (ROS) generation overwhelms the cellular antioxidant capacity. ROS resulting from normal physiological processes could cause oxidative damage to cellular constituents, such as proteins, lipids, and DNA, leading to various cellular dysfunction. Cells are usually protected from oxidative damage by antioxidant enzymes, such as superoxide dismutase (SOD), glutathione peroxidase (GPX), and catalase (CAT), as well as non-enzymatic antioxidants, such as glutathione, vitamin C, and vitamin E [49]. In metabolic syndrome, the balance between ROS and antioxidant could be broken. For instance, the increased oxidation of excess FFA by mitochondrion via beta-oxidation or oxidation of FFA-derived acetyl CoA through the tricarboxylic acid cycle (TCA) cycle generates electron donors, such as nicotinamide adenine dinucleotide and dihydroflavine-adenine dinucleotide, which can cause oxidative stress [50]. FFA, especially palmitate, stimulates diacylglycerol synthesis and activates protein kinase C (PKC), which activates nicotinamide adenine dinucleotide phosphate (NADPH) oxidase, which converts molecular oxygen to its superoxide radicals. The persistent release of FFA from overaccumulation of fat may also activate NADPH oxidase locally (within the adipose cell) or remotely (in other cells), which exacerbates ROS production in obese individuals [51,52]. A study also found that postprandial lipoproteins, especially chylomicrons and VLDL, generate oxygen radicals on the endothelial surface, which reacts with NO and decrease its bioavailability [53,54]. This event will subsequently lead to vascular dysfunction. Therefore, oxidative stress is involved in the pathogenesis of atherosclerosis, hypertension, and T2DM related to metabolic syndrome [55].

Metabolic syndrome is also associated with a state of low-grade inflammation characterised by increased production of cytokines and activation of inflammatory signalling pathways [56]. Chronic oxidative stress transpiring in the adipose tissue may be the first trigger of inflammation, leading to propagation of metabolic syndrome [56]. Obesity is related to a concurrent increase in ROS and expression of NADPH oxidase, and a decrease in the expression of antioxidant enzymes, which are associated with altered adiponectin, IL-6, and monocyte chemoattractant protein (MCP-1) production [57]. Adipose tissue is known to express various adipokines, such as TNF-α, IL-6, IL-1β, leptin, adiponectin, and MCP-1 [58,59]. TNF-α was reported to be high in the adipose tissue of obese individuals, and its level was correlated positively with insulin resistance as it interferes with the insulin signalling transduction via serine-phosphorylation of IRS-1 [60,61]. Other cytokines that are also associated with obesity and insulin resistance are IL-6 and IL-1β [62,63]. These cytokines upregulate obesity and play an important role in the pathophysiological processes underlying metabolic syndrome, T2DM, and CVD [56]. Furthermore, the proinflammatory transcription factor, nuclear factor-kappa B (NF-κB), is also associated with obesity, insulin resistance, and low-grade inflammation [64]. NF-κB regulates the expression of genes that control the expression of inflammatory gene production [65]. Therefore, together with TNF-α and IL-6, its most important activators, NF-κB contributes to metabolic syndrome manifestations [66,67].

The vicious cycle of oxidative stress and inflammation in metabolic syndrome drives the progression of this condition. A natural compound or mixture with both antioxidant and anti-inflammatory properties could break this cycle and relieve metabolic syndrome.

## 3. The Mechanisms of Honey in Reversing Metabolic Changes

Given the role of inflammation and oxidative stress in the development of metabolic syndrome, honey, with the capacity to quench these processes, could prevent metabolic syndrome. The therapeutic effects of honey large depend on the antioxidant and anti-inflammatory properties of its polyphenol and flavonoid content. The phenolic acid and flavonoid profiles of honey vary based on factors, such as climate, geographic factors, and floral abundance [68]. A study on the total phenolic contents and colour intensity of different types of Malaysian honey reported a positive correlation between total phenolic content (TPC) and colour intensity of honey, whereby Kelulut honey had the highest value of TPC of 784.3 mg GAE/kg and other honey, such as Tualang, Pineapple and Borneo, had TPC values of 589.2, 602.4, and 510.4 mg GAE/kg, respectively [69]. The polyphenols and flavonoids are potent antioxidants because they can donate hydrogen and hydrogen groups to scavenge free radicals in oxidative stress. For example, quercetin, caffeic acid (CA), and chlorogenic acid are polyphenols that possess iron-chelating and iron-stabilizing properties, which prevents free radical formation, making them great antioxidants [70,71,72]. Furthermore, polyphenols, such as apigenin, quercetin, and kaempferol, may also exert their anti-inflammatory properties by modulation of enzymes involved in proinflammatory activities, such as nuclear factor-kappa B (NF-κB), activator protein-1 (AP-1), or nuclear factor erythroid 2-related factor 2 (Nrf2) [73,74,75]. These bioactive compounds may contribute synergistically to the antimetabolic effects of honey.

### 3.1. Anti-Inflammatory and Antioxidant Properties of Honey

As illustrated in the previous section, metabolic syndrome is closely linked with oxidative stress and inflammation. Studies have shown that honey can protect against the activation of NF-κB, the key transcription factor of inflammation. An in vitro study reported that 5–20% manuka honey inhibited the activation of NF-κB and AP-1 in *H. pylori*-induced NF-κB and AP-1 DNA-binding activity in gastric epithelial cells and downregulated the expression of cyclooxygenase-2 (COX-2) [76]. Hussein and colleagues also reported the suppression of NF-κB (p65 and p50 gene expression) by Gelam honey at 1.0 and 2.0 g/kg body weight for 7 days in a Carrageenan-induced paw oedema model in rats. Gelam honey at 1 or 2 g/kg also inhibited the nuclear transcription of NF-κB, followed by a subsequent reduction in COX-2 and TNFα [77]. Another animal study by Aziz and colleagues also observed a significant decrease in NF-κB, IL-1β, and TNFα expression, as well as a significant increase in the antioxidant CAT after 28 days of treatment with 1 and 2 g/kg body weight SBH in diabetic rats [14].

On the other hand, honey could activate the nuclear localization of Nrf2, which is the key regulator of the cellular antioxidant defence. Activation of Nrf2, in turn, facilitates the transcription of several Nrf2 target genes that control antioxidant defence and autophagy. Honey activates AMPK and endogenous enzymatic antioxidants, such as SOD, CAT, and GPX [78]. Manuka honey was reported to prevent oxidative damage and preserve mitochondrial functionality via activation of the AMPK/Nrf2 signalling pathway, with a subsequent increase in the expression of antioxidant enzymes, such as SOD and CAT [79]. Another study by Ranneh and colleagues reported that SBH suppressed lipopolysaccharide (LPS)-induced chronic subclinical systemic inflammation (CSSI) and oxidative stress in rats. SBH also reduced NF-κB, p65, and p38 MAPK and upregulated Nrf2 expression in the liver, kidney, heart, and lungs [80]. A study by Sabitha and colleagues reported that p-CA suppressed ethanol-induced oxidative stress and apoptosis by suppressing CYP2E1 and stimulating Nrf2 and its target protein expression in rat liver tissue. Therefore, p-CA is an effective antioxidant by enhancing Nrf2 signalling [81]. Furthermore, honey is also known to reduce malondialdehyde (MDA) levels, which is a product of lipid peroxidation with high reactivity and toxicity, making it one of the most reliable biomarkers of oxidative stress [82]. A study found that supplementation of Tualang honey 1 g/kg for 12 weeks in spontaneously hypertensive rats reduced malondialdehyde (MDA) levels and downregulated the activity of glutathione-S-transferase (GST) and CAT, while it moderately upregulated Nrf2 mRNA expression [16].

### 3.2. Antiobesity Properties of Honey

An in vitro study on Pineapple honey showed a significantly reduced lipid droplet size by 33.78% to 70.36% and reduced lipid accumulation in treated 3T3-L1 adipocytes, suggesting honey might limit the storage of lipids in adipocytes [83]. Malaysian Gelam and Acacia honey were reported to reduce weight gain and BMI in rats with obesity induced by a high-fat diet (HFD). Rats that were fed Gelam honey ad libitum for 4 weeks also showed a reduction in the adiposity index compared to the HFD group, showing Gelam honey can prevent excessive adipose tissue formation [84]. Similar results were demonstrated by Rafie and colleagues using stingless bee honey (SBH) from *Heterotrigona itama*. The rats showed reduced BMI, percentage body weight gain, adiposity index, and relative liver weight after a 6-week honey supplementation (500–1000 mg/kg) [85]. Ramli and colleagues also observed a reduction in body fat and omental fat mass in rats supplemented with 1 g/kg/day of Kelulut honey for 8 weeks [86]. Romero-Silva and colleagues observed a significantly lower weight gain as well as smaller fat cells in rats supplemented with a hypercaloric diet containing 20% honey (unknown source) compared with the sucrose-fed rats for 8 weeks [87].

In addition, human interventional studies also found a similar finding marked by a reduction in body weight, body fat, and lipid profile after consumption of 70 g of natural unprocessed honey collected from Iran for 30 days [88]. Furthermore, a randomised open-labelled controlled clinical study by Pai and colleagues reported a significant reduction in body weight, BMI, WC, hip circumference, and lipid profile in obese patients treated with unprocessed and processed honey (collected from India). The honey was supplemented at 48 g for 48 days in these patients [89].

Honey is rich in polyphenols known to reduce body weight and fat mass, thus explaining its antiobesity properties. A study by Liao and colleague reported that a reduction in body weight as well as a decrease in the ratio of various adipose tissue mass (epididymal, retroperitoneal, and mesenteric fat) and body weight in mice given a high-fat diet with 0.02% and 0.08% *w/w* CA for 6 weeks [90]. A similar study with the supplementation of 50 µg/day of quercetin for 8 weeks in mice fed a high-fat diet also reported a reduction in fat mass and body weight [91]. Another animal study reported CA and chlorogenic acid supplementation (0.02% *w/w*) for 8 weeks in mice fed a high-fat diet significantly reduced BW by 8% and 16%, respectively. Furthermore, the weight of the epididymal white adipose tissue of mice supplemented with CA and chlorogenic acid was lower than the control group by 22% and 46%, respectively. In this experiment, supplementation with CA and chlorogenic acid also reduced plasma leptin, indicating the alleviation of leptin resistance [92]. Meanwhile, an in vitro study concerning the effects of GA on lipolysis found that GA (250 µM 48 and 72 h) inhibits proliferation but induces apoptosis in 3T3-L1 preadipocytes. As evidence, cells treated with 50 µM GA for 12 h showed an increase in Fas (CD95)/Fas Ligand (FasL; CD95L) and p53 expression. These proteins are involved in the extracellular pathway in apoptotic signalling [93]. However, the actions of these polyphenols might be different compared to honey because the content and their interactions with other compounds need to be considered.

Besides, fructo-oligosaccharides (FOSs) are also known to alter lipid metabolism. FOSs in honey are resistant to human digestive enzymes and could act as probiotics. Various studies reported that the administration of these fructans promotes a reduction in weight gain and energy intake in rats [94]. A study from Kaume et al. observed that dietary supplementation of 5% FOSs (*w/w*) could significantly lower total lipids by 12% with a subsequent reduction in liver weight in obese Zucker rats [95]. Daubioul and colleagues also observed that the body weight of FOS-fed Zucker rats was significantly lower than control rats after 4 weeks of treatment. A histological examination of the liver revealed a reduction in fat cells of the rats fed with 10% FOS (*w/w*). Besides, FOS also decreases malic enzyme (ME) activity, a lipogenic key enzyme in providing NADPH for fatty acid elongation by FAS. Therefore, FOS can inhibit the synthesis of long-chain fatty acid [96]. Similarly, Agheli and colleagues reported a reduction of FAS activity after FOS supplementation in insulin-resistant Sprague-Dawley rats [97]. Delzane and Kok showed a reduction in the activity of lipogenic enzymes, such as acetyl-CoA carboxylase (ACC), ME, and FAS after FOS supplementation by modifying the gene expression of these enzymes. Of note, FAS mRNA was reduced by 40% in FOS-fed rats compared to the control group [98].

A summary of the antiobesity properties of honey is presented in Figure 1.

### 3.3. Antihyperglycaemic Properties of Honey

Honey is a sweet substance with a relatively low glycaemic index, making it a suitable sugar substitute. The fructose present in honey contributes to its sweet taste. The major source of fructose used by the food industry as a sweetener is derived from cane sugar or high-fructose corn syrup. Fructose is a potent and acute regulator of liver glucose uptake and glycogen synthesis. It forms TG more effectively and is more lipogenic than glucose despite having a similar chemical structure [99]. In the liver, fructose bypasses the regular steps of glycolysis, catalyzed by glucokinase or hexokinase and phosphofructokinase. Instead, fructose is transported by insulin-independent glucose transporter (GLUT-5) and is metabolized to fructose-1-phosphate by the enzyme fructokinase or ketohexokinase [100]. High fructose intake results in postprandial hypertriglyceridaemia and an increase in visceral adipose deposition. This event exacerbates hepatic triglyceride accumulation, protein kinase C activation, and hepatic insulin resistance due to the continuous portal delivery of fatty acid to the liver [39]. On the other hand, honey can normalise circulating glucose levels because its fructose content can prolong gastric emptying and lowers food intake [78]. The slow absorption of fructose within the intestinal tracks might prolong interaction between fructose and the intestinal receptor, which might result in satiety [18].

The hypoglycaemic properties of honey have been illustrated in rodent models of diabetes, healthy subjects, and diabetic patients. These effects might be contributed by the components of honey, such as fructose, and phenolic acids. A study reported that administration of 1.0 to 2.0 g/kg of Nigerian honey for 3 weeks significantly reduced hyperglycaemia in alloxan-induced diabetic rats [101]. Another animal study also reported similar findings with 1.0 and 2.0 g/kg body weight of SBH from *Geniotrigona thoracica* in diabetic rat models by suppressing FBG levels after 28 days of treatment. Additionally, histopathological changes, expression of oxidative stress, inflammation, and apoptosis markers within the pancreatic islet were improved in conjunction with the increase in the expression of insulin in the islet [14].

In a human intervention study, 30 days of 70 g of honey (collected from Iran) was reported to reduce FBG compared to overweight individuals fed with sucrose [88]. In obese girls, supplementation of 15 g honey for 6 months caused a reduction in BMI and the area under the concentration–time curve (AUC) in an oral glucose tolerance test and insulin [102]. Agrawal and colleagues observed a higher degree of tolerance to honey with a significantly lower glucose level in patients with diabetes or impaired glucose tolerance after consumption of 90 g of unprocessed natural honey from India in 300 mL of water, at a 30-min interval up to 2 h [103]. A case-control study in Egypt involving children and adolescents with type 1 diabetes mellitus also recorded a lower glycaemic index and incremental index in both the diabetic and control group after honey (unspecified source) consumption at a calculated dose (dose in g = subjects’ body weight in kg × 1.75 with a maximum of 75 g) diluted in 200 mL of water, every 30 min postprandial for 2 h [104].

Honey could modulate the key components of the insulin signalling pathway, P13k/Akt [78]. The development of insulin resistance is characterised by an increase in NF-κB, MAPK, and IRS-1 serine phosphorylation. However, pretreatment of oxidative stress-induced HIT-T15 cells with Gelam honey extract (20, 40, 60, and 80 µg/mL) and quercetin (20, 40, 60, and 80 µM) for 24 h, prior to stimulation with 20 and 50 mM glucose, reported an increase in the expression of Akt and decreased expression of IRS-1 serine phosphorylation, NF-κB, and MAPK [105]. In addition, a study by Tapia and colleagues also observed lower serum glucose in rats fed with 10% honey ad libitum for 4 months. Interestingly, long-term honey consumption in the presence of a high-fat diet did not significantly increase the insulin concentration. Honey was reported as the sweetener that increased phosphorylation of IRS-tyrosine and Akt and lowered the protein abundance of NF-κB, which indicates better insulin signalling. It was also observed that honey significantly increased white adipose tissue GLUT-4 expression in rats, an insulin-sensitive glucose transporter that moves glucose into the adipose tissue, indicating better insulin sensitivity [106]. A summary of the antihyperglycaemic properties of honey is presented in Figure 2.

### 3.4. Antihyperlipidaemic Properties of Honey

Many studies on the hypolipidaemic effects of honey have been conducted. A study by Samat et al. reported that the consumption of Gelam and Acacia honey (dose unspecified) for four weeks reduced TG and cholesterol in rats fed a high-fat diet (HFD) [84]. In another report, supplementation of SBH from *Geniotrigona thoracica* honey 1.0 and 2.0 g/kg body weight for 28 days could increase the HDL-C level but reduce TG, TC, and LDL-C levels in streptozotocin-nicotinamide-induced diabetic male rats [14]. This beneficial effect was replicated in human studies, whereby the consumption of 70 g of natural unprocessed honey dissolved in 250 mL of tap water for 30 days in overweight and obese individuals caused a 3.3%, 4.3%, and 19% reduction in TC, LDL-C, and TG [88].

The natural compounds in honey contribute to its lipid-lowering effects. Polyphenols, such as CA and p-coumaric acid (P-CA), commonly seen in all honey possess numerous bioactive properties including antioxidant, anti-inflammatory, and lipid-lowering actions [18]. Studies on CA and p-coumaric acid have shown that both compounds can reduce the mRNA expression of SREBP-1c and FAS and inhibit their activity [90,107].

FFA is the main player in the synthesis of TG in hepatocytes. In metabolic syndrome, lipolysis in adipose tissue increases, resulting in enhanced FFA delivery to the liver [108]. Both SREBP-1c and FAS are key regulators to FFA synthesis, and their dysregulation is the primary source of hypertriglyceridaemia [109]. In recent studies, phenolic compounds can activate 5′adenosine monophosphate-activated protein kinase (AMPK), which mediates the reduction in SREBP-1c protein expression by preventing SREBP-1c nuclear translocation, subsequently suppressing FAS synthase expression [107,110].

According to Liao and colleagues, CA reduced the TG and cholesterol content in oleic acid-induced hepatic lipogenesis in HepG2. Furthermore, CA also enhanced the phosphorylation of AMPK and ACC, which are lipid oxidation-related proteins. CA also downregulated the lipogenesis gene expression of SREBP-1 and its target gene FAS in the presence of oleic acid [90].

Gallic acid (GA) and catechin can also be found in honey. Chang and colleagues’ study showed that oleic acid significantly increased FAS, SREBP-1, and phosphorylated AMPK expression in high-fat diet-treated mice. An extract containing GA and catechin reduced FAS expression by 6% and SREBP-1 by 23%. This observation indicated that GA and catechin might attenuate hepatic lipid accumulation by regulating FA and TG synthesis [111]. Another study by Kim and colleagues demonstrated that p-CA (up to 40 µg) increased the phosphorylation of AMPK and ACC, and the expression of carnitine palmitoyltransferase-1a in HepG2 cells, suggesting enhanced fatty acid β-oxidation. Furthermore, p-CA also reduced lipid accumulation in HepG2 cells, implying its ability to attenuate FA synthesis. The authors also implied that p-CA might inhibit the lipid uptake of HepG2 cells [107]. The effects of these individual compounds might not fully recapitulate the antihyperlipidaemic potential of honey, as they might act in synergy with other components to achieve the overall effects. A summary of the antihyperlipidaemic properties of honey is presented in Figure 3.

### 3.5. Antihypertensive Properties of Honey

Flavonoids present in honey, such as quercetin and kaempferol, show promising results in the treatment of cardiovascular diseases [112]. A study by Sanchez and colleagues reported that treatment with 10 mg/kg of quercetin for 13 weeks lowered blood pressure and heart rate in spontaneously hypertensive rats. This was achieved by upregulating eNOS and p47 protein expression and reducing NADPH-oxidase-mediated superoxide anion generation, which attenuated endothelial dysfunction [113]. Another in vitro study showed that quercetin (>0.1 µM) increased the conductance of calcium-activated potassium channel (BKca) currents in rat coronary artery rings. Given that BKca regulates coronary artery tone in vivo, quercetin could induce vasodilatation [114]. Kuhlmann and colleagues reported quercetin improves endothelial dysfunction by inducing BKca-dependant endothelial hyperpolarisation in human endothelial cells derived from the umbilical cord vein (HUVEC) (with maximum effect achieved at 50 µM/L). This event led to an influx of extracellular calcium ions, resulting in increased NO production [115]. Similarly, an in vitro study demonstrated that quercetin exerted vasodilatory effects on the human umbilical artery. In a double-blind randomised placebo-controlled study, healthy volunteers were given capsules containing placebo, 200, or 400 mg quercetin randomly in 3 consecutive weeks. The result showed that quercetin increased the brachial diameter [116].

In addition, quercetin also reduced endothelial proliferation by 56% and increased cyclic guanosine monophosphate (cGMP) level by 5 fold due to NO formation [115]. An in vitro study by Shen and colleagues indicated that quercetin improved endothelial dysfunction in the mouse abdominal aorta and aortic ring. At a dose of 5 and 10 µM, quercetin significantly increased acetylcholine-mediated endothelial-dependant relaxation in the presence of hypochlorous acid (HOCl) by 34% and 78%, respectively. Incubation with 10 µM quercetin 2 h prior to treatment with HOCl also restored eNOS activity in aortic tissue. Quercetin activated eNOS activity, subsequently increasing NO production [117].

Meanwhile, pre-treatment of kaempferol in HUVEC was reported to suppress the expression of NF-κB by LPS significantly. In addition, TNFα was increased in LPS-stimulated endothelial cells, which was significantly decreased with kaempferol. Altogether, the results suggest that kaempferol improves barrier integrity, and inhibits the activity of cell adhesion and migration to endothelial cells by inhibiting NF-κB expression and TNFα production, thereby promoting its benefits in the treatment of vascular inflammatory disease [118]. The individual effects of these bioactive compounds in honey and their interactions contribute to its overall antihypertensive effects. A summary of the antihypertensive properties of honey is presented in Figure 4.

## 4. Conclusions

Honey is a potential agent in reversing metabolic syndrome through its antiobesity, hypoglycaemic, hypolipidaemic, and hypotensive actions. These properties are exerted through the components in honey, like polyphenols, which act as potential lipogenic enzyme inhibitors. Through synergistic actions, these polyphenols can limit weight gain and adipose tissue formation. The antioxidant and anti-inflammatory effects of these polyphenols also prevent endothelial dysfunction and ultimately, hypertension. Honey is shown to improve insulin sensitivity and normalize glucose metabolism despite its carbohydrate content. In conclusion, honey can be used as an adjuvant therapy for prevention of metabolic syndrome in general, by mechanisms such as reducing oxidative stress and inflammation. Thus, it a beneficial food substance that can be incorporated for the prevention and management of metabolic syndrome.

## Figures and Tables

**Figure 1 molecules-26-00808-f001:**
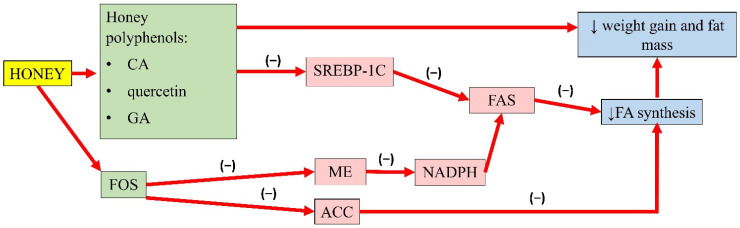
Summary of the antiobesity properties of honey. CA, caffeic acid; GA, gallic acid; FOS, fructo-oligosaccharides; SREBP-1C, sterol regulatory element-binding transcription factor 1C; ME, malic enzyme; ACC, acetyl-CoA carboxylase; FAS, fatty acid synthase; NADPH, nicotinamide adenine dinucleotide phosphate; FA, fatty acid; (−), inhibit; ↓, reduce.

**Figure 2 molecules-26-00808-f002:**
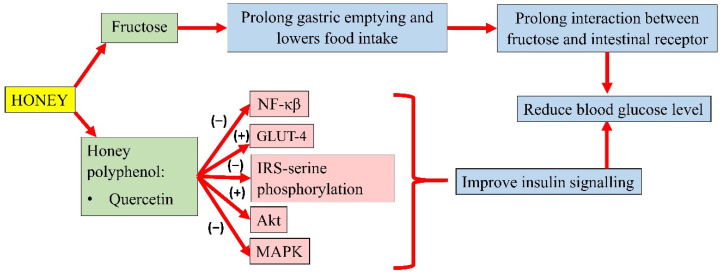
Summary of the antihyperglycemic properties of honey. NF-κB, nuclear factor kappa B; GLUT-4, insulin-independent glucose transporter 4; IRS-serine phosphorylation, inhibiting insulin receptor substrate-serine phosphorylation; Akt, protein kinase B; MAPK, mitogen-activated protein kinase; (−), inhibit; (+), increase.

**Figure 3 molecules-26-00808-f003:**
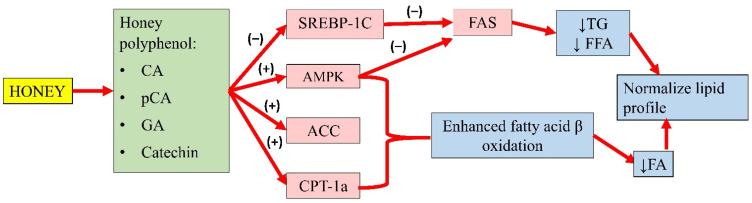
Summary of the antihyperlipidaemic properties of honey. CA, caffeic acid; p-CA, p-coumaric acid; GA, gallic acid; SREBP-1c, sterol regulatory element-binding transcription factor 1c; AMPK, 5′adenosine monophosphate-activated protein kinase; ACC, acetyl-CoA carboxylase; CPT-1a, carnitine palmitoyltransferase-1a; FAS, fatty acid synthase; TG, serum triglyceride; FFA, free fatty acid; FA, fatty acid; (−), inhibit; (+), increase; ↓, reduce.

**Figure 4 molecules-26-00808-f004:**
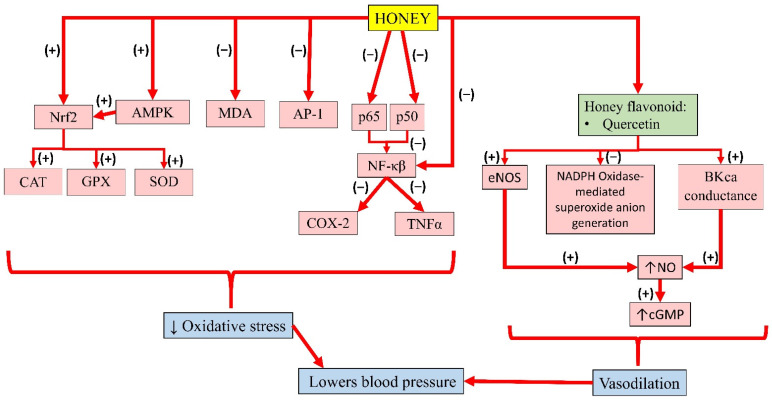
Summary of the antihypertensive properties of honey. Nrf2, nuclear factor erythroid 2-related factor 2; AMPK, 5′adenosine monophosphate-activated protein kinase; MDA, malondialdehyde; AP-1, activator protein-1; p65, subunit 65 protein of NF-κB; p50, subunit 50 protein of NF-κB; NF-κB, nuclear factor kappa B; CAT, catalase; GPX, glutathione peroxidase; SOD, superoxide dismutase; COX-2, cyclooxygenase-2; TNFα, tumour necrosis factor-alpha; eNOS, endothelial nitric oxide synthase; NADPH, nicotinamide adenine dinucleotide phosphate; BKca, large-conductance calcium-activated potassium channel; NO, nitric oxide; cGMP, cyclic guanosine monophosphate; (−), inhibit; (+), increase; ↓, reduce; ↑, increase.

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
