# Peer review of "The Mechanism of Honey in Reversing Metabolic Syndrome"

_molecules, 2021, doi:10.3390/molecules26040808_

Round 1

Reviewer 1 Report

This paper would be the continuation of a previous work, collecting on this occasion evidence on the possible mechanisms by which honey would act interfering in lipid metabolism, modification of oxidative stress levels, and reduction of blood pressure. While it's an interesting review, there's still a lot to know to be able to recommend honey to prevent or help in SM.  It would be interesting if they particularly adjusted the conclusion.

Conclusion: Honey can be an interesting food as functional food or adjuvant to prevention of Metabolic Syndrome in general, such as reducing oxidative stress, hypertension and promoting weight gain situations. The point I would change would be the conclusion and give strength to all points where honey can possibly intervene by producing metabolic modifications.

Author Response

Dear reviewer,

Thank you for reviewing our manuscript. We appreciate the constructive comments provided and have responded to each of them in the response sheet. Changes in the text are highlighted in yellow.

Thank you again for reviewing our manuscript. We hope that the revised manuscript can fulfil the required standard of the esteemed journal. We look forward to receiving your positive reply.

Reviewer 2 Report

Review of manuscript ref. molecules-1088866; title: "The Mechanism of Honey in Reversing Metabolic Syndrome", by Hashim et al.

General comment: the study reviews the chemo-protective effect of honey and main honey compounds on the metabolic syndrome. The authors state the main pathophysiological landmarks of metabolic syndrome and then specifically focus on the particular effects of honey components on the molecular pathways involved in the syndrome. Updated revisions of bioactive products and compounds from nature that are present in our diet should be always welcome to improve our knowledge and understanding of the healing processes and nutritional prevention of diseases. This is a very interesting review dealing with the preventing and reverting effects of honey in a condition of metabolic syndrome. The idea is sound, writing and organization are clear, and data is mostly recent. A major concern is that some beneficial effects of honey are only reported for specific compounds and not for the whole product. The authors should limit the purposed beneficial effects of honey to its main phenolic components, which should be clearly indicated in an introductory epigraph. Some specific comments are detailed below:

Specific comments:

  • Line 185; statement: in rodents, lipogenesis is an important source of VLDL and TG formation, seems unnecessary since the pathway is the most important source of VLDL and TG in at least all mammals. Besides, the paragraph and the whole text of epigraph 2.3 is not focused on rodent research. The sentence seems to be out of context.
  • Lines 219-225; the authors should support the text including one or two generic references on oxidative stress.
  • Figure 2, it should say IRS-serine phosphorylation.
  • Line 402; it should be mL, with capital L.
  • Lines 409-415; the paragraph is not really focused on the effects of honey; it is very generic and quoted references 88-90 do not seem to deal with phenolic compounds and 91 deals with a rare compound which might not even be within the composition of honey.
  • Lines 442-476; since the text within these lines deals mainly inflammation and antioxidant defenses perhaps the authors should consider renaming the epigraph 3.4 as anti-inflammatory and antioxidant effects of honey and start a new epigraph 3.5 in line 477 as anti-hypertensive effects. The authors directly correlate oxidative stress as the unbalance of redox status with endothelial dysfunction and hypertension, but oxidative stress is also related to type 2 diabetes and dyslipidemia among many other diseases, therefore, it should not be strictly linked to hypertension; perhaps anti-inflammatory and antioxidant effects deserve a particular chapter or epigraph. The same comment applies to point 2 on pathophysiology of metabolic syndrome.
  • Line 475; the meaning of MDA as a biomarker of lipid peroxidation and oxidative damage should be shortly explained.
  • Lines 504-510; kaempferol is not indicated at top of page 6 as one of the main phenolic components of honey. The fact that many phenolic compounds show biological effects does not presume that same effect in honey.

Author Response

(The authors gave the same response as above.)

Round 2

Reviewer 2 Report

General comment: the authors have conveniently addressed al my comments and queries, thus I recommend accepting the revised version 1 for publication at Molecules.

Specific comments:

  • There is still a minor error in Figure 2, it should say IRS-serine phosphorylation, not IRS-serene.